# Residual Pathway Priors for Soft Equivariance Constraints

**Marc Finzi**[*]
New York University

**Greg Benton**[*]
New York University

**Andrew Gordon Wilson**
New York University

## Abstract

Models such as convolutional neural networks restrict the hypothesis space to a set of functions satisfying equivariance constraints, and improve generalization in problems by capturing relevant symmetries. However, symmetries are often only partially respected, preventing models with restriction biases from fitting the data. We introduce Residual Pathway Priors (RPPs) as a method for converting hard architectural constraints into soft priors, guiding models towards structured solutions while retaining the ability to capture additional complexity. RPPs are resilient to approximate or misspecified symmetries, and are as effective as fully constrained models even when symmetries are exact. We show that RPPs provide compelling performance on both model-free and model-based reinforcement learning problems, where contact forces and directional rewards violate the assumptions of equivariant networks. Finally, we demonstrate that RPPs have broad applicability, including dynamical systems, regression, and classification.

## 1 Introduction

Central to the expanding application of deep learning to structured data like images, text, audio, sets, graphs, point clouds, and dynamical systems, has been a search for finding the appropriate set of inductive biases to match the model to the data. These inductive biases, such as recurrence [43], local connectivity [29], equivariance [10], or differential equations [8], reduce the set of explored hypotheses and improve generalization. Equivariance in particular has had a large impact as it allows ruling out a large class of meaningless shortcut features in many distinct domains, such as the ordering of the nodes in graphs and sets or the coordinate system chosen for an image.

A disadvantage of hard coding these restrictions is that this prior knowledge may not match reality. A scene may have long range non-local interactions, rotation equivariance may be violated by a preferred camera angle, or a dynamical system may occasionally have discontinuous transitions. In particular, symmetries are delicate. A small perturbation like adding wind breaks the rotational symmetry of a pendulum, and bumpy or tilted terrain could break the translation symmetry for locomotion. In these cases we would like to incorporate our prior knowledge in a way that admits our own ignorance, and allows for the possibility that the world is more complex than we imagined. We aim to develop an approach that is more general, and can be applied when symmetries are exact, approximate, or non-existent.

The Bayesian framework provides a mechanism for expressing such knowledge through priors. In much of the past work on Bayesian neural networks, the relationship between the prior distribution and the functions preferred by the prior is not transparent. While it is easy to specify different variances for different channels, or to use heavy tailed distributions, it is hard know how high level properties meaningfully translate into these low level attributes. Ultimately priors should represent our prior *beliefs*, and those beliefs we have are about high level concepts like the locality, independence, and symmetries of the data.

---

[*]Equal Contribution

35th Conference on Neural Information Processing Systems (NeurIPS 2021).

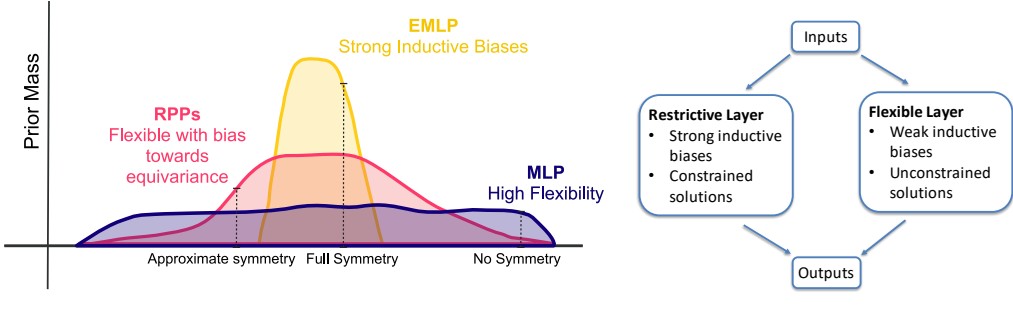

(a) Priors over Equivariant Solutions  (b) Structure of RPP Models

Figure 1: **Left:** RPPs encode an Occam's razor approach to modeling. Highly flexible models like MLPs lack the inductive biases to assign high evidence to key datasets, while models with strict equivariance constraints are not flexible enough to support problems with only approximate symmetry. **Right:** The structure of RPPs. Expanding the layers into a sum of the constrained and unconstrained solutions while setting the prior to favor the constrained solution, leads to the more flexible layer explaining only the *residual* of what is already explained by the constrained layer.

To address the need for more interpretable priors we introduce *Residual Pathway Priors* (RPPs), a method for converting hard architectural constraints into soft priors. Practically, RPPs allow us to tackle problems in which perfect symmetry has been violated, but approximate symmetry is still present, as is the case for most real world physical systems.

RPPs have a prior bias towards equivariant solutions, but are not constrained to equivariance. The choice of RPPs can be viewed from an Occam's razor perspective, shown in Figure 1a, in which we seek to use models that have both the correct inductive biases and level of flexibility [41]. As we find with problems in which symmetries exist, highly flexible models with weak inductive biases like MLPs fail to concentrate prior mass around solutions that exhibit any symmetry. On the other hand when symmetries are only approximate, the strong biases of constrained models like Equivariant Multi-Layer Perceptrons (EMLP) [14] fail to provide support for the observations. As a middle ground between these two extremes, RPPs combine the inductive biases of constrained models with the flexibility of MLPs to define a model class which excels when data show approximate symmetries.

In the following sections we introduce our method and show results across a variety of domains. We list our contributions and the accompanying sections below:

1. We propose *Residual Pathway Priors* as a mechanism to imbue models with soft inductive biases, without constraining flexibility.
2. While our approach is general, we use RPPs to show how to turn hard architectural constraints into soft equivariance priors (Section 4).
3. We demonstrate that RPPs are robust to varying degrees of symmetry (Section 5). RPPs perform well under exact, approximate, or misspecified symmetries.
4. Using RPP on the approximate symmetries in the complex state spaces of the Mujoco locomotion tasks, we improve the performance of model free RL agents (Section 6).

We provide a PyTorch implementation of residual pathway priors at
`https://github.com/mfinzi/residual-pathway-priors`.

## 2   Related Work

The challenge of equivariant models not being able to fully fit the data has been identified in a number of different contexts, and with different application specific adjustments to mitigate the problem. Liu et al. [32] observe that convolutional networks can be extremely poor at tasks that require identifying or outputting spatial locations in an image as a result of the translation symmetry. The authors solve the problem by concatenating a coordinate grid to the input of the convolution layer. Constructing translation and rotation equivariant GCNNs, Weiler and Cesa [53] find that in order to get the best performance on CIFAR-10 and STL-10 datasets which have a preferred camera orientation, they

must break the symmetry which they do by using equivariance to progressively smaller subgroups in the later layers. Bogatskiy et al. [6] go to great lengths to construct Lorentz group equivariant networks for tagging collisions in particle colliders only to break it by introducing dummy inputs that identify the collision axis. van der Wilk et al. [50] use the marginal likelihood to learn approximate invariances in Gaussian processes from data. In a similar approach, Benton et al. [5] explore a related concept of learning the *extent* of the symmetry using the reparametrization trick and test time augmentation. While sharing some commonalities with RPP, this method is not aimed at achieving approximate equivariance and cannot bake equivariance into the model architecture.

Outside of equivariance, adding model outputs to a much more restrictive base model has been a fruitful idea employed in multiple contexts. The original ResNet [20, 21] drew on this motivation, with shortcut connections. Johannink et al. [24], Silver et al. [45] proposed Residual Reinforcement Learning, whereby the RL problem is split into a user designed controller using engineering principles and a flexible neural network policy learned with RL. Similarly, in modeling dynamical systems, one approach is to incorporate a base parametric form informed by models from physics or biology, and only learn a neural network to fit the delta between the simple model and reality [25, 33].

There have been several works tackling symmetries and equivariance in RL, such as permutation equivariance for multi agent RL [46, 23, 31], as well exploring reflection symmetry for continuous control tasks [1], and discrete symmetries in the more general framework of MDP homomorphisms [49]. However, in each of these applications the symmetries need to be exact, and the complexities of real data often require violating those symmetries. Although not constructed with this purpose, some methods which use regularizers to enforce equivariance [48] could be used for approximate symmetries. Interestingly, the value of approximate symmetries of MDPs has been explored in some theoretical work [42, 47], but without architectures that can make use of it. Additionally data augmentation, while not able to bake in architectural equivariance, has had successful application for encouraging equivariance on image tasks [27] and recently even on tabular state vectors [30, 36].

## 3 Background

In order develop our method, we first review the concept of group symmetries, how representations formalize the way these symmetries act on different objects.

**Group Symmetries** In the machine learning context, a symmetry group $G$ can be understood as a set of invertible transformations under which an object is the same, such as reflections or rotations. These symmetries can act on many different kinds of objects. A rotation could act on a simple vector, a 2d array like an image, a complex collection objects like the state space of a robot, or more abstractly on an entire classification problem or Markov Decision Process (MDP).

**Representations** The way that symmetries act on objects is described by a *representation*. Given an object in an $n$-dimensional vector space $V$, a group representation is a mapping $\rho : G \to \mathbb{R}^{n \times n}$, yielding a matrix which acts on $V$. Vectors $v \in V$ are transformed $v \mapsto \rho(g)v$. In deep learning, each of the inputs and outputs to our models can be embedded in some vector space: an $m \times m$ sized rgb image exists in $\mathbb{R}^{3m^2}$, and a node valued function on a graph of $m$ elements exists within $\mathbb{R}^m$. The representation $\rho$ specifies how each of these objects transform under the symmetry group $G$.

These representations can be composed of multiple simpler subrepresentations, describing how each object within collection transform. For example given the representation $\rho_1$ of rotations acting on a vector in $\mathbb{R}^3$ and a representation $\rho_2$ of how rotations act on a $3 \times 3$ matrix, the two objects concatenated together have a representation given by $\rho_1(g) \oplus \rho_2(g) = \begin{bmatrix} \rho_1(g) & 0 \\ 0 & \rho_2(g) \end{bmatrix}$ where the two matrices are concatenated along the diagonal. Practically this means we can represent intricate and multifaceted structures by breaking them down into their component parts and defining how each part transforms. For example, we may know that the velocity vector, an orientation quaternion, a joint angle, and a control torque all transform in different ways under a left-right reflection, and one can accommodate this information into the representation.

**Equivariance** Given some data $X$ with representation $\rho_{\text{in}}$ and $Y$ with representation $\rho_{\text{out}}$, we may wish to learn some mapping $f : X \to Y$. A model $f$ is equivariant [10], if applying the symmetry

transformation to the input is equivalent to applying it to the output

$$f(\rho_{\text{in}}(g)x) = \rho_{\text{out}}(g)f(x).$$

In other words, it is not the symmetry of $X$ or $Y$ that is relevant, but the symmetry of the function $f$ mapping from $X$ to $Y$. If the true relationship in the data has a symmetry, then constraining the hypothesis space to functions $f$ that also have the symmetry makes learning easier and improves generalization [12]. Equivariant models have been developed for a wide variety of symmetries and data types like images [10, 54, 56, 53], sets [55, 35], graphs [34], point clouds [3, 15, 44], dynamical systems [13], jets [6], and other objects [51, 14].

## 4   Residual Pathway Priors

In this section, we introduce Residual Pathway Priors. The core implementation of RPP is to expand each layer in model into a sum of both a restrictive layer that encodes the hard architectural constraints and a generic more flexible layer but penalize the more flexible path via a lower prior probability. Through the difference in prior probability, explanations of the data using only the constrained solutions are prioritized by the model; however, if the data is more complex the residual between the target and the constrained layer will be explained using the flexible layer. We can apply this procedure to any restriction priors, such as linearity, locality, markovian structure, and of course equivariance.

The Residual Pathway Prior draws inspiration from the residual connections in ResNets [20, 21], whereby training stability and generalization improves by providing multiple paths for gradients to flow through the network that have different properties. One way of interpreting a residual block and shortcut connection $f(x) = x + h(x)$ in combination with l2 regularization, either explicitly from weight decay or implicitly from the training dynamics [39], is as a prior that places higher prior likelihood on the much simpler identity mapping than on the more flexible function $h(x)$. In this way, $h(x)$ need only explain the the difference between what is explained in the previous layer (passed through by $I$) and the target.

We can leverage a similar approach to convert the hard constraints of a restriction prior specified through a given network architecture into a soft prior that merely places higher prior likelihood on such models. Supposing we have an equivariant layer $A(x)$, and a more flexible non-equivariant layer $B(x)$ which contains $A$ as a special case, we can allow the model to explain part of the data with $A$ and part with $B$ by forming the sum $A(x) + B(x)$. A given a prior likelihood on the size of $A$ and $B$ such as $p(A) \propto \exp\left(-\|A\|^2/2\sigma_a^2\right)$ and $p(B) \propto \exp\left(-\|B\|^2/2\sigma_b^2\right)$ with $\sigma_a > \sigma_b$, a MAP optimized model will favor explanations of the data using the more structured layer $A$ and only resort to using layer $B$ to explain the *difference* between the target and what is already explained by the more structured model $A$. Adding these non-equivariant residual pathways to each layer of an equivariant model, we have a model that has the same expressivity of a network formed entirely of $B$ layers, but with the inductive bias towards a model formed entirely with the equivariant $A$ layers. We term this a *residual pathway prior* (RPP).

To make the approach concrete, we first consider constructing equivariance priors using the constraint solving approach known as Equivariant Multi-Layer Perceptrons (EMLP) from Finzi et al. [14].

**Equivariant MLPs**   EMLPs provide a method for automatically constructing exactly equivariant layers for any given group and representation by solving a set of constraints. The way in which the vectors are equivariant is given by a formal specification of the types of the input and output through defining their representations. Given some input vector space $V_{\text{in}}$ with representation $\rho_{\text{in}}$ and some output space $V_{\text{out}}$ with representation $\rho_{\text{out}}$ the space of all equivariant linear layers mapping $V_{\text{in}} \to V_{\text{out}}$ satisfies

$$\forall g \in G: \quad \rho_{\text{out}}(g)W = W\rho_{\text{in}}(g).$$

These solutions to the constraint form a subspace of matrices $\mathbb{R}^{n_{\text{out}} \times n_{\text{in}}}$ which can be solved for and described by a $r$ dimensional orthonormal basis $Q \in \mathbb{R}^{n_{\text{out}} n_{\text{in}} \times r}$. Linear layers can then be parametrized in this equivariant basis. The elements of $W$ can be parametrized $\text{vec}(W) = Q\beta$ for $\beta \in \mathbb{R}^r$ for the linear layer $v \mapsto Wv$, and symmetric biases can be parametrized similarly.

**Equivariance Priors with EMLP**   In order to convert the hard equivariance constraints in EMLP into a soft prior over equivariance that can accomodate approximate symmetries, we can apply the

RPP procedure from above to each these linear layers in the network. Instead of parametrizing the weights $W$ directly in the equivariant basis $\text{vec}(W) = Q\beta$, we can instead define $W$ as the sum $W = A + B$ of an equivariant weight matrix $\text{vec}(A) = Q\beta$ with a Gaussian prior over $\beta$ and an unconstrained weight matrix $B \sim \mathcal{N}(0, \sigma_b^2 I)$ with a Gaussian prior. A Gaussian prior over $\beta \sim \mathcal{N}(0, \sigma_a^2 I)$ is equivalent to the Gaussian prior $A \sim \mathcal{N}(0, \sigma_a^2 QQ^\top)$. Since $Q$ is an orthogonal matrix, we can breakdown the covariance of $B$ into the covariance in the equivariant subspace $Q$ as well as the covariance in the orthogonal complement $P$, $\sigma_b^2 I = \sigma_b^2 QQ^\top + \sigma_b^2 PP^\top$. Therefore the sum of the weight matrices is distributed as $A + B = W \sim \mathcal{N}(0, (\sigma_a^2 + \sigma_b^2)QQ^\top + \sigma_b^2 PP^\top)$.

*Regardless* of the values of the prior variances $\sigma_a^2$ and $\sigma_b^2$, solutions in the equivariant subspace $QQ^\top$ are automatically favored by the model and assigned higher prior probability mass than those in the subspace $PP^\top$ that violate the constraint. Even if $\sigma_b > \sigma_a$, the model still favors equivariance because the equivariance solutions are contained in the more flexible layer $A$. We show in Section 5.2 that RPPs are insensitive to the choice of $\sigma_a$ and $\sigma_b$ provided that $\sigma_a$ is large enough to be able to fit the data. By replacing each of the equivariant linear layers in an EMLP with a sum of an equivariant layer and an unconstrained layer and adding in the negative prior likelihood to the loss function, we arrive at RPP-EMLP that can accommodate approximate or incorrectly specified symmetries. [1]

**RPPs With Other Equivariant Models**  While in EMLP these equivariant bases are solved for explicitly, the argument holds for the linear layers in other equivariant networks in precisely the same way. A good example of this is the translationally equivariant convolutional neural network (CNN) which can be viewed as a restricted subset of a fully connected network. Though the layers are parametrized as convolutions, the convolution operation can be expressed as a Toeplitz matrix residing within the space of dense matrices. Adding the convolution to a fully connected layer and choosing a prior variance $\sigma_a^2$ and $\sigma_b^2$ over each, we have the same RPP prior

$$W \sim N(0, \sigma_a^2 QQ^\top + \sigma_b^2 I) \tag{1}$$

where $Q$ is the basis of (bi-)Toeplitz matrices corresponding to $3 \times 3$ filters. This RPP CNN has the biases of convolution but can readily fit non translationally equivariant data. Likewise for other equivariant models like GCNNs [10], and we can even apply the RPP principle to the breaking of a given symmetry group to a subgroup.

## 5   How and Why RPPs Work

In this section we explore how and why RPPs work on a variety of domains, applying RPPs in settings where constraints are known to be helpful, cannot fully describe the problem, and are misspecified.

### 5.1   Dynamical Systems and Levels of Equivariance

In order to better understand how and why residual pathway priors interact with the symmetries of the problem we move to settings in which we can directly control both the type of symmetry and the level to which the symmetries are violated. We examine how RPPs coupled with EMLP networks perform (RPP-EMLP) on the inertia and double pendulum datasets featured in Finzi et al. [14] in 3 experimental settings: $(i)$ the original inertia and double pendulum datasets which preserve exact symmetries with respect to the to O(3) and O(2) groups respectively; $(ii)$ modified versions of these datasets in which add an additional terms (such as wind on the double pendulum) that lead to approximate symmetries; and $(iii)$ versions with misspecified symmetry groups which break the symmetries entirely (described in Appendix C).

The results for these 3 settings are given in Figure 4. Across all settings RPP-EMLP match the performance of EMLP when symmetries are exact, perform as well as an MLP when the symmetry is misspecified and better than both when the symmetry is approximate. For these experiments we use a prior variance of $\sigma_a^2 = 10^5$ on the EMLP weights and $\sigma_b^2 = 1$ on the MLP weights.

**Exact Symmetries**  As part of the motivation, RPPs should properly attribute prior mass to both constrained and unconstrained solutions, we test cases in which symmetries are exact, and show that

---

[1]For the EMLP that uses gated nonlinearities which do not always reduce to a standard Swish, we likewise add a more general Swish weighted by a parameter with prior variance $\sigma_b^2$.

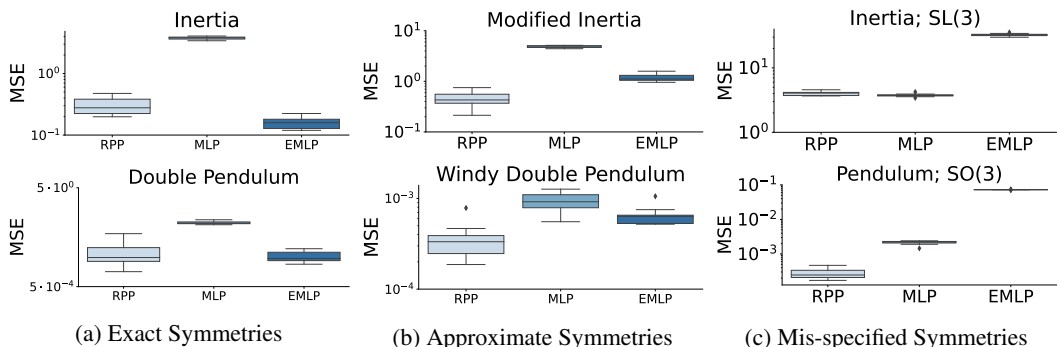

(a) Exact Symmetries      (b) Approximate Symmetries      (c) Mis-specified Symmetries

Figure 2: A comparison of test performance over 10 independent trials using RPP-EMLP and equivalent EMLP and MLP models on the inertia (**top**) and double pendulum (**bottom**) datasets in which we have three varying levels of symmetries. The boxes represent the interquartile range, and the whiskers the remainder of the distribution. **Left:** perfect symmetries in which EMLP and the equivariant components of RPP-EMLP exactly capture the symmetries in the data. **Center:** approximate symmetries in which the perfectly symmetric systems have been modified to include some non-equivariant components. **Right:** mis-specified symmetries in which the symmetric components of EMLP and RPP-EMLP do not reflect the symmetries present in the data.

RPP-EMLP is capable of performing on par with EMLP which only admits solutions with perfect symmetry. The results in Figure 4a show that although the prior over models as described RPP-EMLP is broader than that of EMLP (as we can admit non-equivariant solutions) in the presence of perfectly equivariant data RPP-EMLP do not hinder performance, and we are able to generalize nearly as well as the perfectly prescribed EMLP model.

**Approximate symmetries** To better showcase the ideas of Figure 1 we compare RPP-EMLPs to EMLPs and MLPs on the modified inertia and windy pendulum (Figure **??**) datasets. In these datasets we can think about the systems as primarily equivariant but containing non-equivariant contributions. As shown in 4b these problems are best suited for RPP-EMLP as MLPs have no bias towards the approximately symmetry present in the data, and EMLPs are overly constrained in this setting.

**Misspecified symmetries** In contrast to working with perfect symmetries and showing that RPP-EMLPs are competitive with EMLPs, we also show that when symmetries are *mis*specified the bias towards equivariant solutions does not hinder the performance of RPP-EMLPs. For the inertia dataset we substitute the group equivariance in EMLP from $O(3)$ to the overly large group $SL(3)$ consisting of all volume and orientation preserving linear transformations, not just the orthogonal ones. For the double pendulum dataset, we substitute $O(2)$ symmetry acting on $\mathbb{R}^3$ with the larger $SO(3)$ rotation group that contains it but is not a symmetry of the dataset.

By purposefully misspecifying the symmetry in these datasets we intentionally construct EMLP and RPP-EMLP models with incorrect inductive biases. In this setting EMLP is incapable of making accurate predictions as it has a hard constraint on an incorrect symmetry. Figure 4c shows that even in cases where the model is intentionally mis-specified that RPPs can overcome a poorly aligned inductive bias and recover solutions that perform as well as standard MLPs, even where EMLPs fail.

### 5.2 Prior Levels of Equivariance

To test the effect of prior variances we use the modified inertia dataset, which represents a version of a problem in which perfect equivariance has been broken by adding new external forces to the dynamical system. Shown in Figure 3 (right) is a comparison of mean squared error on test data as a function of the prior precision terms on both the equivariance and basic weights. As a general trend we see that when the regularization on the equivariant weights is too high (equivalent to a concentrated prior around 0) we find instability in test performance, yet when we apply a broad prior to the equivariant weights performance is typically both better in terms of MSE, and more stable to the choice of prior on the basic model weights.

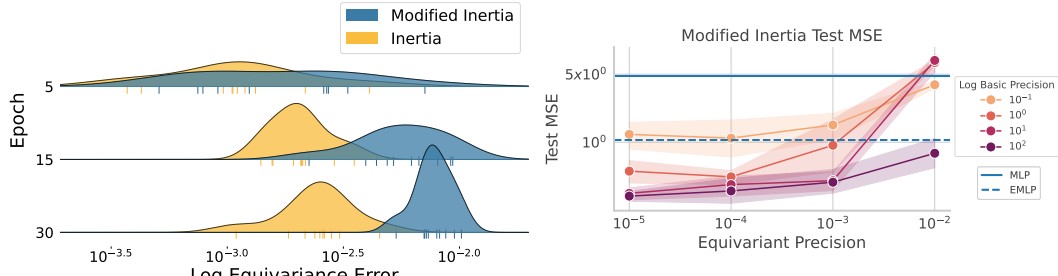

Figure 3: **Left:** Kernel density estimators of log equivariance error across training epochs for $10$ independently trained networks. Here the color denotes the dataset these models were trained on. Treating these samples as a proxy for posterior density, we see that on the non-equivariant Modified Inertia dataset, the posterior is shifted upward to match the level of equivariance in the data during training. **Right:** Test MSE as a function of the weight decay parameters on the equivariant and basic weights on the modified inertia dataset. We observe that so long as the prior in the basis of equivariant weights is broad enough, we can achieve low test error with RPPs.

As the prior variances over the equivariant basis $Q$ and the non-equivariant basis $P$ describe our bias towards or away from equivariant solutions we investigate how the choice of prior variance relates to the level of symmetry present in a given dataset. In the windy pendulum dataset of Figure **??** we have control over the level of wind and thus how far our system is from perfect equivariance.

### 5.3 Posterior Levels of Equivariance

RPPs describe a method for setting a prior over equivariance, and in the presence of new data we expect the posterior distribution over equivariance to change accordingly. Using samples from a deep ensemble to query points of high density in the posterior we estimate how the distribution over equivariance error progresses through training. Recalling that with an equivariant function $f$ we have $\rho_2(g)f(x) = f(\rho_1(g)x)$, we compute equivariance error as

$$\text{EquivErr}(f,x) = \text{RelErr}(\rho_2(g)f(x), f(\rho_1(g)x)) \text{ where } \text{RelErr}(a,b) = \frac{\|a-b\|}{\|a\|+\|b\|}. \quad (2)$$

We train one deep ensemble on the inertia dataset which exhibits perfect symmetry, and another on the modified inertia dataset which has only partial symmetry, with each deep ensemble being comprised of $10$ individual models using the same procedure as in Section 5.1. In Figure 3 (left) we see that throughout training the models trained on the modified inertia concentrate around solutions with substantially higher equivariance error than models trained on the dataset with the exact symmetry. This figure demonstrates one of the core desiderata of RPPs: that we are able to converge to solutions with an appropriate level of equivariance for the data.

### 5.4 RPPs and Convolutional Structure

Using the RPP-Conv specified by the prior in Eqn 1 we apply the model to CIFAR-10 classification and UCI regression tasks where the inputs are reshaped to zero-padded two dimensional arrays and treated as images. Notably, the model is still an MLP and merely has larger prior variance in the convolutional subspace. As a result it can perform well on image datasets where the inductive bias is aligned, as well as on the UCI data despite not being an image dataset as shown in Table 1. While retaining the flexibility of an MLP, the RPP performs better than the locally connected MLPs trained with $\beta$-lasso in Neyshabur [38] which get $14\%$ error on CIFAR-10. The full details for the architectures and training procedure are given in Appendix C.

## 6 Approximate Symmetries in Reinforcement Learning

Both model free and model based reinforcement learning present opportunities to take advantage of structure in the data for predictive power and data efficiency. On the one hand stands the use of

|      | CIFAR-10 | Energy | Fertility | Pendulum | Wine |
|------|----------|--------|-----------|----------|------|
| MLP  | $37.61 \pm 0.56$ | $0.39 \pm 0.48$ | $0.049 \pm 0.0044$ | $4.65 \pm 0.50$ | $0.66 \pm 0.058$ |
| RPP  | $12.62 \pm 0.34$ | $0.73 \pm 0.44$ | $0.060 \pm 0.0097$ | $4.25 \pm 0.50$ | $0.69 \pm 0.031$ |
| Conv | $12.03 \pm 0.46$ | $1.34 \pm 0.38$ | $0.076 \pm 0.0157$ | $4.63 \pm 0.36$ | $0.79 \pm 0.092$ |

Table 1: Mean test classification error on CIFAR-10 and MSE on 4 UCI regression tasks, with one standard deviation errors taken over 10 trials. Similar to Figure 4, we find that whether the constrained convolutional structure is helpful (CIFAR) or not (UCI), RPP-Conv performs similarly to the model with the correct level of complexity.

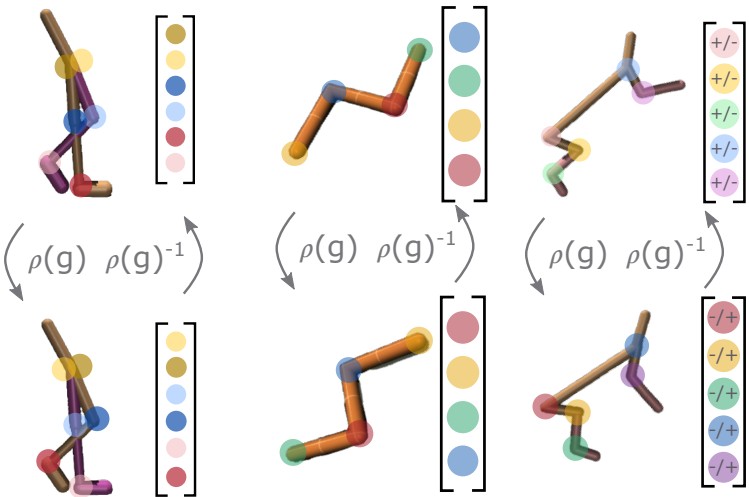

(a) Walker2d Left-Right  (b) Swimmer Front-Back  (c) HalfCheetah In-Out

Figure 4: Example illustrations of symmetries and representations from the Mujoco environments. **Left:** left-right symmetry in the *Walker2d* environment, **center:** front-back symmetry in the *Swimmer* environment, and **right:** In-out similarity in the *HalfCheetah* environment

model predictive control in the engineering community where finely specified dynamics models are constructed by engineers and only a small number of parameters are fit with system identification to determine mass, inertia, joint stiffness, etc. On the other side of things stands the hands off approach taken in the RL community, where general and unstructured neural networks are used for both transition models [9, 52, 22] as well as policies and value functions [17]. The state and action spaces for these systems are highly complex with many diverse inputs like quaternions, joint angles, forces, torques that each transform in different ways under a symmetry transformation like a left-right reflection or a rotation. As a result, most RL methods treat these spaces a black box ignoring all of this structure, and as a result they tend to require tremendous amounts of training data, making it difficult to apply to real systems without the use of simulators.

We can make use of this information about what kinds of objects populate the state and action spaces to encode approximate symmetries of the RL environments. As shown in van der Pol et al. [49], exploiting symmetries in MDPs by using equivariant networks can yield substantial improvements in data efficiency. But symmetries are brittle, and minor effects like rewards for moving in one direction, gravity, or even perturbations like wind, a minor tilt angle in CartPole, or other environment imperfections can break otherwise perfectly good symmetries. As shown in Table 2, broadening the scope to approximate symmetries allows for leveraging a lot more structure in the data which we can exploit with RPP. While Walker2d, Swimmer, Ant, and Humanoid have exact left/right reflection symmetries, Hopper, HalfCheetah, and Swimmer have approximate front/back reflection symmetries. Ant and Humanoid have an even more diverse set, with the $D_4$ dihedral symmetry by reflecting and cyclicly permuting the legs of the ant, as well as continuous rotations of the Ant and Humanoid within the environment which can be broken by external forces or rewards. Identifying this structure

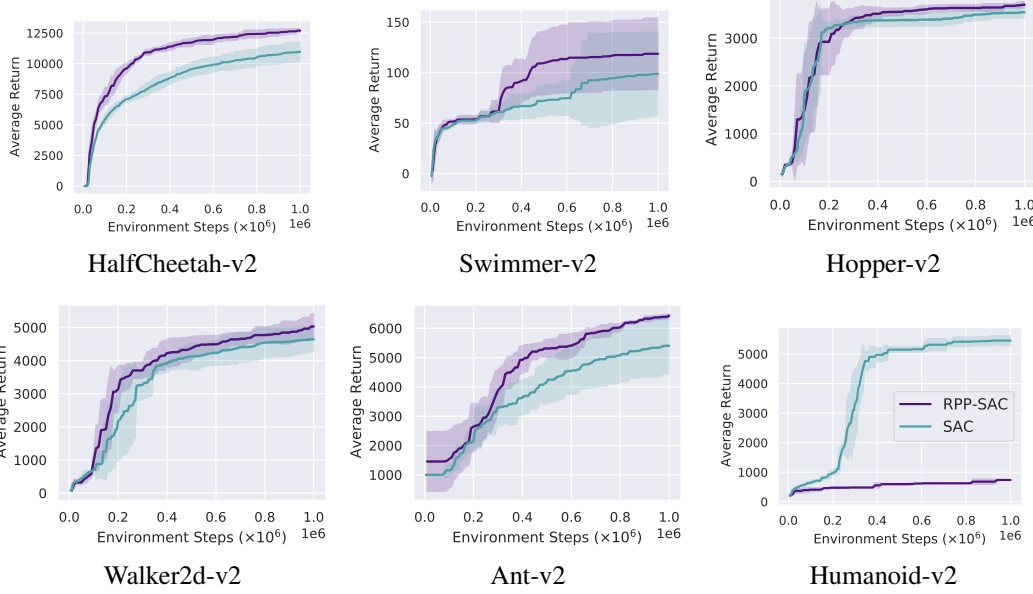

Figure 5: Average reward curve of RPP-SAC and SAC trained on Mujoco locomotion environments (max average reward attained at each step). Mean and one standard deviation taken over $4$ trials shown in the shaded region. Incorporating approximate symmetries in the environments improves the efficiency of the model free RL agents.

in the data, we are able to use the generality of EMLP to construct an equivariant model for this data, and then turn it into a soft prior using RPP.

| Symmetries | Walker2d | Hopper | HalfCheetah | Swimmer | Ant | Humanoid |
|------------|----------|--------|-------------|---------|-----|----------|
| Exact | $\mathbb{Z}_2$ | ✗ | ✗ | $\mathbb{Z}_2$ | $\mathbb{Z}_2$ | $\mathbb{Z}_2$ |
| Approximate | $\mathbb{Z}_2$ | $\mathbb{Z}_2$ | $\mathbb{Z}_2$ | $\mathbb{Z}_2 \times \mathbb{Z}_2$ | $D_4 \times O(2)$ | $\mathbb{Z}_2 \times O(2)$ |
| This work | $\mathbb{Z}_2$ | $\mathbb{Z}_2$ | $\mathbb{Z}_2$ | $\mathbb{Z}_2 \times \mathbb{Z}_2$ | $\mathbb{Z}_4$ | $SO(2)$ |

Table 2: Exact and approximate symmetries of Mujoco locomotion environments of which we use the subgroups in the bottom row, see Appendix D for the detailed action and state representations.

## 6.1 Approximate Symmetries in Model Free Reinforcement Learning

We evaluate RPPs on the standard suite of Mujoco continuous control tasks in the context of model-free reinforcement learning. With the appropriately specified action and state representations detailed in Appendix D, we construct RPP-EMLPs which we use as a drop-in replacement for both the policy and Q-function in the Soft Actor Critic (SAC) algorithm [17], using the same number of layers and channels. In contrast with van der Pol et al. [49] where equivariance is used just for policies, we find that using RPP-EMLP for the policy function alone is not very helpful with Actor Critic (see Figure 5). With the exception of the Humanoid-v2 environment where the RPP-EMLP destabilizes SAC, we find that incorporating the exact and approximate equivariance with RPP yields consistent improvements in the data efficiency of the RL agent as shown in Figure 5.

## 6.2 Better Transition Models for Model Based Reinforcement Learning

We also investigate whether the equivariance prior of RPP can improve the quality of the predictions for transition models in the context of model based RL. To evaluate this in a way decoupled from the complex interactions between policy, model, and value function in MBRL, we instead construct a static dataset of $50,000$ state transitions sampled uniformly from the replay buffer of a trained SAC

|          | Swimmer-v2 | | Hopper-v2 | | Ant-v2 | |
|----------|------------|------------|-----------|-----------|-----------|-----------|
| Rollout  | MLP        | **RPP**    | MLP       | **RPP**   | **MLP**   | RPP       |
| 10 Steps   | $0.51 \pm 0.02$ | $\mathbf{0.40 \pm 0.04}$ | $1.1 \pm 0.1$ | $\mathbf{0.9 \pm 0.1}$ | $\mathbf{4.2 \pm 0.1}$ | $5.2 \pm 0.3$ |
| 30 Steps   | $1.6 \pm 0.2$ | $\mathbf{1.26 \pm 0.14}$ | $3.8 \pm 0.3$ | $\mathbf{3.1 \pm 0.5}$ | $\mathbf{11.3 \pm 0.2}$ | $13.9 \pm 0.7$ |
| 100 Steps  | $3.9 \pm 1.0$ | $\mathbf{2.75 \pm 0.31}$ | $9.8 \pm 0.5$ | $\mathbf{7.0 \pm 0.7}$ | $\mathbf{16.0 \pm 0.3}$ | $20.0 \pm 1.1$ |
| Equiv Err  | 46%        | 19%        | 98%       | 32%       | 36%       | 31%       |

Table 3: Transition model rollout relative error in percent % averaged over 10, 30, and 100 step rollouts (geometric mean over trajectory). Errorbars are 1 standard deviation taken over 3 random seeds. Equivariance error is computed from as the geometric mean averaged over the 100 step rollout.

agent. Since the trajectories in the replay buffer come from different times, they capture the varied dynamics MBRL transition models often encounter during training.

State of the art model based approaches on Mujoco tend to use an ensemble of small MLPs that predict the state transitions [9, 52, 22, 2], without exploiting any structure of the state space. We evaluate test rollout predictions via the relative error of the state over different length horizons for the RPP model against an MLP, the method of choice. As shown in Table 3, RPP transition models outperform MLPs on the Swimmer and Hopper environments, especially for long rollouts showing promise for use in MBRL. On these environments, RPP learns a smaller but non-negligible equivariance error that still enables it to fit the data.

# 7   Limitations

Using RPP-EMLP for the state and action spaces of the Mujoco environments required identifying the meaning of each of the components in terms of whether they are scalars, velocity vectors, joint angles, or orientation quaternions, and also which part of the robot they correspond to. This can be an error-prone process. While RPPs are fairly robust to such mistakes, the need to identify components makes using RPP more challenging than standard MLP. Additionally, due to the bilinear layers within EMLP, the Lipschitz constant of the network is unbounded which can lead to training instabilities when the inputs are not well normalized. We hypothesize these factors may contribute to the training instability we experienced using RPP-EMLP on Humanoid-v2.

# 8   Conclusion

In this work we have presented a method for converting restriction priors such as equivariance constraints into flexible models that have a bias towards structure but are not constrained by it. Given uncertainty about the nature of the data, RPPs are a safe choice. These RPP models are able to perform as well as the equivariant models when exact symmetries are present, and as well as unstructured MLPs when the specified symmetry doesn't exist, and better than both for approximate symmetries. We have shown that encoding approximate symmetries can be a powerful technique for improving performance with the messy and complex state and action spaces in reinforcement learning, as well as in a more general regression setting.

We hope that RPP enables designing more expressive priors for neural networks that capture the kinds of high level assumptions that machine learning researchers actually hold when developing models, rather than low level characteristics about the parameters that are hard to interpret. Building better techniques for enforcing high level properties helps lower the cost of incorporating prior knowledge, and better accommodate the complexities of data even if they don't match our expectations.

**Acknowledgements** We thank Samuel Stanton for useful discussion and feedback. This research was supported by an Amazon Research Award, NSF I-DISRE 193471, NIH R01DA048764-01A1, NSF IIS-1910266, and NSF 1922658NRT-HDR: FUTURE Foundations, Translation, and Responsibility for Data Science.

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
