# Residual Pathway Priors for Soft Equivariance Constraints
## Supplementary Material

## Appendix Outline

In Section 7 discuss potential for negative impact. In Section B we investigate the utility of using RPP-EMLP for the policy function only on the Mujoco tasks. In Section C we detail the datasets and experimental methodology used in the paper. Finally in Sections D and E we break down the components of the Mujoco environment state and action spaces, and the representations that we use for them.

## A    Potential Negative Impacts

As one of our primary application areas is reinforcement learning, and specifically exploiting approximate symmetries in reinforcement learning, we must address the potential negative impacts of the deployment of RPPs in RL systems. In general model free RL algorithms tend to be brittle, and often policies and behavior learned in a simulated environment like Mujoco don't transfer easily to real world robots. This point is acknowledged by most RL researchers, and a large effort is being made to improve the situation. Applying neural networks to the control of real robots can be dangerous if the functions are important or failure can cause injury to the robot or humans. We believe that RL will ultimately be impactful for robot control, however practitioners need to be responsible and exercise caution.

## B    Benefit of Equivariant Value Functions

In principle both the policy and the value or critic function can benefit from equivariance. However, the policy learns from the value function in the policy update which is approximately equivalent to minimizing the KL divergence

$$\mathbb{E}_{s \sim \mathcal{D}}[\mathrm{KL}(\pi_\phi(\cdot|s) \,|\, \exp(Q_\theta(\cdot, s))/Z_\theta(s))]$$

as derived in Haarnoja et al. [18]. If the value function $Q$ is a standard MLP yielding a non equivariant distribution and the policy function $\pi$ is an RPP that merely has a bias towards equivariance, then the RPP policy will learn to fit the non equivariant parts of $Q$ as if it were a ground truth dataset that is not equivariant. This likely explains why we find in practice that using an RPP for the value function has a stronger impact on performance as shown in Figure 5.

## C    Experimental Details

Here we present the training details of the models used in the paper. Experiments were run on private servers with NVIDIA Titan RTX and RTX 2080 Ti GPUs. We estimate that all runs performed in the initial experimentation and final evaluation on the RL tasks used approximately 500 GPU hours. The experiments on dynamical systems, CIFAR-10, and UCI data required an additional 200 GPU hours.

### C.1    Synthetic Dataset Experiments (5.1 and 5.3)

The windy pendulum dataset is a variant of the double spring pendulum Hamiltonian system from Finzi et al. [14]. In addition to the Hamiltonian of the base system

$$H_0(x_1, x_2, p_1, p_2) = V(x_1, x_2) + T(p_1, p_2)$$

where $T(p_1, p_2) = \|p_1\|^2/2m_1 + \|p_2\|^2/2m_2$ and $V(x_1, x_2) =$

$$\tfrac{1}{2}k_1(\|x_1\| - \ell_1)^2 + \tfrac{1}{2}k_2(\|x_1 - x_2\| - \ell_2)^2 + m_1 g^\top x_1 + m_2 g^\top x_2,$$

we add a perturbation $H_1(x_1, x_2, p_1, p_2) = -w^\top x_1 - w^\top x_2$ that is the energy of the wind acting as a constant force pushing in the $w = [-8, -5, 0]$ direction. Setting $H = H_0 + \epsilon H_1$, we can control the strength of the wind and we choose $\epsilon = 0.01$. This perturbation breaks the $SO(2)$ symmetry about the $z$ axis.

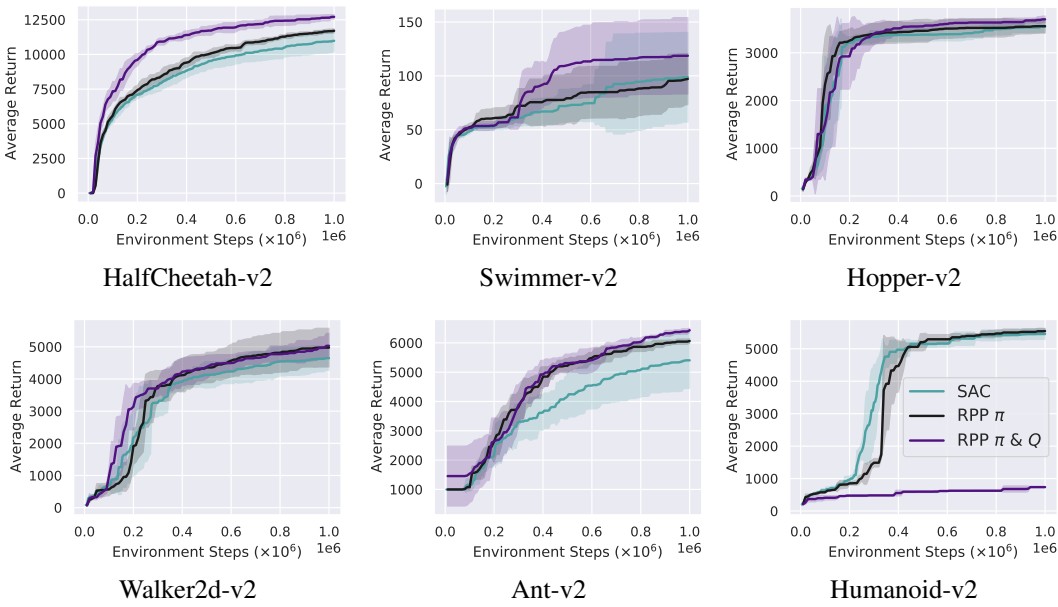

Figure 6: Average reward curves (max over steps) for an RPP-EMLP applied to the policy $\pi$ only, as well as an RPP-EMLP for both the policy $\pi$ and the critic $Q$. Mean and standard deviation taken over 4 trials shown in the shaded region. Only minor performance gains are achieved if using RPP for the policy only, however this variant is more stable and can to train on Humanoid-v2 without diverging.

For the MLP, EMLP, and RPP we use 3 layer deep 128 hidden unit Hamiltonian neural networks [16] to fit the data using the rollouts of an ODE integrator [8] with an MSE loss on rollouts of length 5 timesteps with $\Delta t = 0.2$. For training we use 500 trajectory chunks and use another 500 for testing. We train all models in section 5.1 for 1000 epochs, sufficient for convergence. The input and output representation for EMLP and RPP-EMLP is $V_{O(3)}^4 \to \mathbb{R}$, where $V_{O(3)}$ is the restricted representation from the standard representation of a 3D rotation matrix to the given group in question, like $SO(2)$ for rotations about the $z$ axis. The input is $V_{O(3)}^4$ because there are two point masses each of which has a 3D vectors for position and for momentum. The scalar $\mathbb{R}$ output is the Hamiltonian function.

The Modified Inertia dataset is a small regression dataset off of the task also from Finzi et al. [14] for learning the moment of inertia matrix in 3D of a collection of 5 point masses. For the base Inertia dataset, the targets are $\mathcal{I} = \sum_{i=1}^{5} m_i (x_i^\top x_i I - x_i x_i^\top)$ from the input tuples $(m_i, x_i)_{i=1}^5$. In order to break the equivariance of the dataset, we add an additional term so that the target is $y = \text{vec}(\mathcal{I} + 0.3 \mathcal{I}^2 \hat{z} \hat{z}^\top \mathcal{I})$ where $\hat{z}$ is the unit vector along the $z$ axis. The input and output representations for EMLP and RPP-EMLP on this problem are $(\mathbb{R} \oplus V)^5 \to V \otimes V$, representing the 5 point masses and vectors mapping to matrices $V \otimes V$.

We use 1000 train and test examples for the inertia datasets and we train for 500 epochs. In both cases we use an Adam optimizer [26] with a learning rate of $0.003$.

## C.2 Image and UCI experiments (5.4)

We use the CIFAR-10 and UCI datasets, taken from Krizhevsky et al. [28] and Dua and Graff [11] respectively. In Section 5 we train models on dynamical systems and CIFAR-10 and UCI regression data. For the CIFAR-10 experiments we use a convolutional neural network (and the equivalent MLP) with 9 convolutional layers and 1 fully connected layer, and max-pooling layers after the third and sixth convolutional layers. The channel sizes of the 9 layers are, in order: $16, 16, 16, 32, 32, 32, 32, 32, 32$. We train for 200 epochs using a cosine learning rate schedule with an initial learning rate of $0.05$ and the Adam optimizer.

For the UCI tasks we use a small convolutional neural network, and the equivalent MLP, with 3 convolutional layers and 1 fully connected layer, with each convolutional layer having 32 channels.

Models are trained for 1000 epochs using an Adam optimizer with a learning rate of 0.01 and a cosine learning rate schedule.

## C.3  Model Free RL

We train on the Mujoco locomotion tasks in the OpenAI gym environments [7]. We follow the implementation details and hyperparameters from Haarnoja et al. [19], with a learned temperature function, stochastic policies, and double critics. Additionally we use the recommendation from Andrychowicz et al. [4] to initialize the last layer of the policy network with 100x smaller weights, which we find slightly improves the performance of both RPP and the baseline. Additionally for RPP which can be less stable than standard SAC, we use the Adam betas $\beta_1 = 0.5$ and $\beta_2 = 0.999$ that are used in he GAN community [37] rather than the defaults. Training with the RPP $\pi$ and $Q$ functions on the Mujoco locomotion tasks takes about 8 hours for 1 million steps.

We found it necessary to reduce the speed $\tau$ of the critic moving average to keep SAC stable on some of the environments, with values shown in Table 4. In general, higher $\tau$'s are favorable for learning quickly. Unfortunately we were not able to get SAC with an RPP Q function to train reliably on Humanoid, even after trying multiple values of $\tau$.

| | Walker2d | Hopper | HalfCheetah | Swimmer | Ant | Humanoid |
|---|---|---|---|---|---|---|
| Baseline $\tau$ | .005 | .005 | .005 | .005 | .005 | .005 |
| RPP $\tau$ | .004 | .005 | .005 | .004 | .005 | ✗ |

Table 4: Critic moving average speed $\tau$.

## C.4  Transition Models for Mujoco

We train the transition models on a dataset of 50000 transitions which are composed of 5000 trajectory chunks of length 10. These trajectory chunks are sampled uniformly from the replay buffer collected over the course of training a standard SAC agent for $10^6$ steps on each of the environments. We train by minimizing the $\ell 1$ norm of the rollout error over a 10 step trajectory, and we evaluate on a holdout set of 50 trajectories of length 100.

The models are simple MLPs or RPPs mapping from the state and control actions to the state space, predicting the change in state,

$$x_{t+1} = x_t + \mathrm{NN}(x_t, u_t).$$

For the MLPs and RPPs we use 2 hidden layers of size 256 as well as swish activations [40]. We use a prior variance of $10^6$ in the equivariant subspace and 3 in the non equivariant subspace. The RPP is a standard RPP-EMLP with the input representation $\rho_X \oplus \rho_U$ (concatenation of the representation of the state space and the action space), output representation $\rho_X$, and symmetry group described in Appendix D the same as for the model free experiments. We train the transition models for 500 epochs which takes about 45 minutes for RPP compared to 15 minutes for the standard MLPs.

## D  Mujoco State and Action Representations

Based on the state and action spaces of the Mujoco environments we describe in Appendix E, we define appropriate group representations on these spaces. Let $V$ be the base representation of the group acted upon by permutations for $\mathbb{Z}_n$ and by rotation matrices for $\mathrm{SO}(2)$, let $\mathbb{R}$ denote a scalar representation (of dimension 1) that is unaffected by the transformations, and let $P$ be a pseudoscalar representation (of dimension 1) that transforms by the sign of the permutation. For $\mathbb{Z}_2$, $P$ takes the values 1 and $-1$ and acts by negating the values when a flip or L/R reflection is applied.

From the raw state and action spaces listed in Appendix E, we convert quaternions to 3D rotation matrices for Humanoid and Ant, and we reorder elements to group together left/right pairs for Walker2d and Swimmer. The representations of these transformed state and action vectors are shown in Table 5. Note that $V^3$ denotes $V \oplus V \oplus V = V^{\oplus 3}$, and is simply the concatenation of 3 copies of $V$ as $\mathbb{R}^3$ would be 3 copies of $\mathbb{R}$. This is not to be confused with powers of the tensor product, $V^{\otimes 3} = V \otimes V \otimes V$. For Humanoid, we denote the restricted representation of 3D rotation matrices restricted to the $\mathrm{SO}(2)$ rotations about the $z$ axis as $V_{\mathrm{SO}(3)}$.

Table 5: Mujoco Locomotion State and Action Representations used for RPP-EMLP

| Env | State Representation | Action Rep | Group |
|---|---|---|---|
| Hopper | $\mathbb{R} \oplus P^5 \oplus \mathbb{R} \oplus P^4$ | $P^3$ | $\mathbb{Z}_2$ |
| Swimmer | $\mathbb{R} \oplus P_\leftrightarrow \oplus (P_\leftrightarrow \otimes V_\updownarrow) \oplus (\mathbb{R} \oplus P)^2 \oplus (P_\leftrightarrow \otimes V_\updownarrow)$ | $P_\leftrightarrow \otimes V_\updownarrow$ | $\mathbb{Z}_2^\leftrightarrow \times \mathbb{Z}_2^\updownarrow$ |
| HalfCheetah | $\mathbb{R} \oplus P^8 \oplus \mathbb{R} \oplus P^7$ | $P^6$ | $\mathbb{Z}_2$ |
| Walker2d | $\mathbb{R}^2 \oplus V^3 \oplus \mathbb{R}^3 \oplus V^3$ | $V^3$ | $\mathbb{Z}_2$ |
| Ant | $\mathbb{R}^5 \oplus V^2 \oplus \mathbb{R}^6 \oplus V^2$ | $V^2$ | $\mathbb{Z}_4$ |
| Humanoid | $\mathbb{R} \oplus V_{\text{SO(3)}}^{\otimes 2} \oplus \mathbb{R}^{17} \oplus V_{\text{SO(3)}}^2 \oplus \mathbb{R}^{17}$ | $\mathbb{R}^{17}$ | SO(2) |

# E   Mujoco State and Action Spaces

In order to build symmetries into the state and action representations for Mujoco environments, we need to have a detailed understanding of what the state and action spaces for these environments represent. As these spaces are not well documented, for each of the Mujoco environments we experimented in the simulator and identified the meanings of the state vectors in Tables 10, 12, 11, 7, 9, 6, and 8. We hope that these detailed descriptions can be useful to other researchers.

Table 6: Hopper-v2 State and Action Spaces

| | |
|---|---|
| | X (Unobserved) |
| | Y |
| | Orientation Angle |
| | Hip Angle |
| | Knee Angle |
| | Ankle Angle |
| State Space | X Velocity |
| | Y Velocity |
| | Orientation Angular Velocity |
| | Hip Angular Velocity |
| | Knee Angular Velocity |
| | Ankle Angular Velocity |
| | Hip |
| Action Space | Knee |
| | Ankle |

Table 7: Swimmer-v2 State and Action Spaces

| | |
|---|---|
| | X (Unobserved) |
| | Y (Unobserved) |
| | Orientation Angle |
| | Head Joint Angle |
| | Tail Joint Angle |
| State Space | X Velocity |
| | Y Velocity |
| | Orientation Angular Velocity |
| | Head Joint Angular Velocity |
| | Tail Joint Angular Velocity |
| Action Space | Head Joint |
| | Tail Joint |

| | Table 8: HalfCheetah-v2 State and Action Spaces | | Table 9: Walker2d-v2 State and Action Spaces |
|---|---|---|---|

| | | | |
|---|---|---|---|
| State Space | X (Unobserved) | State Space | X (Unobserved) |
| | Y | | Y |
| | Orientation Angle | | Orientation Angle |
| | Rear Hip Angle | | Right Hip Angle |
| | Rear Knee Angle | | Right Knee Angle |
| | Rear Ankle Angle | | Right Ankle Angle |
| | Front Hip Angle | | Left Hip Angle |
| | Front Knee Angle | | Left Knee Angle |
| | Front Ankle Angle | | Left Ankle Angle |
| | X Velocity | | X Velocity |
| | Y Velocity | | Y Velocity |
| | Orientation Angular Velocity | | Orientation Angular Velocity |
| | Rear Hip Angular Velocity | | Right Hip Angular Velocity |
| | Rear Knee Angular Velocity | | Right Knee Angular Velocity |
| | Rear Ankle Angular Velocity | | Right Ankle Angular Velocity |
| | Front Hip Angular Velocity | | Left Hip Angular Velocity |
| | Front Knee Angular Velocity | | Left Knee Angular Velocity |
| | Front Ankle Angular Velocity | | Left Ankle Angular Velocity |
| Action Space | Rear Hip | Action Space | Right Hip |
| | Rear Knee | | Right Knee |
| | Rear Ankle | | Right Ankle |
| | Front Hip | | Left Hip |
| | Front Knee | | Left Knee |
| | Front Ankle | | Left Ankle |

Table 10: Ant-v2 State and Action Spaces

| | |
|---|---|
| | X (Unobserved) |
| | Y (Unobserved) |
| | Z |
| | Orientation Quaternion ($4D$) |
| | Limb 2 Left/Right |
| | Limb 2 Up/Down |
| State Space | Limb 3 Left/Right |
| | Limb 3 Up/Down |
| | Limb 4 Left/Right |
| | Limb 4 Up/Down |
| | Limb 1 Left/Right |
| | Limb 1 Up/Down |
| | Limb 1 Left/Right |
| | Limb 1 Up/Down |
| | Limb 2 Left/Right |
| | Limb 2 Up/Down |
| Action Space | Limb 3 Left/Right |
| | Limb 3 Up/Down |
| | Limb 4 Left/Right |
| | Limb 4 Up/Down |

Table 11: Humanoid-v2 Action Space

| | |
|---|---|
| | Torso Forward/Backward |
| | Torso Z |
| | Torso Left/Right |
| | Right Hip Left/Right |
| | Right Hip Up/Down |
| | Right Hip Front/Back |
| | Right Knee Front/Back |
| | Left Hip Left/Right |
| | Left Hip Up/Down |
| Action Space | Left Hip Front/Back |
| | Left Knee Front/Back |
| | Right Shoulder Left/Right |
| | Right Shoulder Front/Back |
| | Right Elbow Front/Back |
| | Left Shoulder Left/Right |
| | Left Shoulder Front/Back |
| | Left Elbow Front/Back |

Table 12: Humanoid-v2 State Space

| | |
|---|---|
| | X (Unobserved) |
| | Y (Unobserved) |
| | Z |
| | Orientation Quaternion ($4D$) |
| | Torso Z |
| | Torso Forward/Backward |
| | Torso Left/Right |
| | Right Hip Left/Right |
| | Right Knee Left/Right |
| | Right Hip Up/Down |
| | Right Knee Up/Down |
| | Left Hip Left/Right |
| State Space (Position) | Left Knee Left/Right |
| | Left Hip Up/Down |
| | Left Knee Up/Down |
| | Right Shoulder Left/Right |
| | Right Shoulder Up/Down |
| | Right Elbow Left/Right |
| | Left Shoulder Left/Right |
| | Left Shoulder Up/Down |
| | Left Elbow Left/Right |
| | Body Linear Velocity ($3D$) |
| | Body Angular Velocity ($3D$) |
| | Torso Z |
| | Torso Forward/Backward |
| | Torso Left/Right |
| | Right Hip Left/Right |
| | Right Knee Left/Right |
| | Right Hip Up/Down |
| | Right Knee Up/Down |
| | Left Hip Left/Right |
| State Space (Velocity) | Left Knee Left/Right |
| | Left Hip Up/Down |
| | Left Knee Up/Down |
| | Right Shoulder Left/Right |
| | Right Shoulder Up/Down |
| | Right Elbow Left/Right |
| | Left Shoulder Left/Right |
| | Left Shoulder Up/Down |
| | Left Elbow Left/Right |