# OpenReview forum: "Residual Pathway Priors for Soft Equivariance Constraints"
_NeurIPS.cc/2021/Conference — NeurIPS 2021 Poster_

### Official Review · Reviewer_hyfb · 2021-07-15

**Rating:** 7
**Confidence:** 3

**Summary:**

This paper proposes a method to introduce inductive biases into neural networks in a *soft* way. In a nutshell, it achieves this by combining strict equivariant layers with flexible non-structured layers, akin to how residual layers combine a identity skip connection with a normal linear layer. This induces soft equivariance priors that, in principle, can accomodate both exact and approximate symmetries.

In a first set of experiments, the paper demonstrates that the proposed RPPs provide a trade-off between unstructured MLPs and hard-equivariant EMLP's in datasets with exact, approximate, and mis-specified symmetries. Then, it showcases the promise of RPPs in the context of reinforcement learning, where recent deep learning-based models have typically been fully unstructured, and for which the results show RPPs provide significant improvements.

**Limitations And Societal Impact:**

Yes, albeit tangentially, in the appendix

**Main Review:**

This paper contains one simple main idea (soft equivariance via residual-type additive layers), but which is motivated very compellingly, executed elegantly, and validated convincingly. It is also written well, and makes for an entertaining read. Overall, I appreciate this type of simple-but-well-executed papers, and I think it makes for a valuale contribution to the conference. Most of my comments and concerns relate to clarity, minor issues, and some requests for additional results.

Meta-Comments:
* Originality. As far as I can tell, this approach to softly enforce equivariance constraints through residual pathway priors is new, although admittedly I'm not on top of all the recent equivariance literature
* Significance. While the experimental results are somewhat limited to make a definitive conclusion on the applicability, robustness, and performance of RPP-based models, they are promising, making this paper potentially quite significant
* Quality. In terms of quality, albeit simple and limited, the theoretical derivations are correct, and the experimental results seem reliable and replicable
* Clarity. The clarity of the paper can be improved in some aspects, e.g., see comments below.

Specific Comments:
* I found it counterintuitive that the solutions for the modified inertia setting in Fig 3 are less spread than those of the pure Inertia ones. Given the approximate notion of symmetry encoded in this dataset, I would have expected it to lead to solutions with more variance in their equivariance error. I would appreciate if the authors could comment on this.
* In Sec 5.3, it would be nice if the main text could specifiy whether the exact same architecture is used for all these datasets. If so, how much padding is there on the UCI datasets?
* Table 2 needs some more explanation. E.g., it is my understanding that the Hooper dataset does have exact symmetries, so it's not clear why it's crossed out for 'Exact' in this table.
* The results in Table 3 and Figure 5 consider only a fully-unstructured MLP baseline. In order to get a full picture here, it would have been nice to have results for some hard-equivariance baseline.
* While I appreciate the discussion on limitations in the Appendix, deferring it to the Supplement instead of the main text dilutes its perceived importance, and goes against the spirit of the 9-th page policy of this year's CFP

Questions:
* In L135, is the prior on the *size*, or on the *magnitude*?
* In L168, do you mean *less* flexible layer? (or replace A by B), otherwise this seems to contradict the intuition for A/B provided above
* Why does the method fail for the Humanoid-v2 setting in Figure 5? The paper says it 'destabilitizes SAC'. How so? Could this be prevented? How prevalent is this phenomenon? Answering such questions would bring better understanding of the limitations of this class of models.


Typos:
* word in missing in L67
* coma missing in L67
* repeated 'the' L129
* The definition of the space of solutions in L150.5 would be clearer wrapping it around {$ W \in \mathbb{R}^{n_{\text{out}} \times n_{\text{in}}} : \quad \dots $}.
* break<->down L162
* A couple of typos in L194
* The figure reference in L197 is incorrect

**Time Spent Reviewing:**

5 hours

---

> ### Author Response · Authors · 2021-08-10
> **Author Response**
>
> Thank you for your constructive comments and thoughtful and supportive review! To address your comments:
> - Regarding Figure 3, one component of the dispersion that is seen between the posterior densities of the model is an artifact of being plotted in log equivariance error, for example a density dispersed from -3 to -4 in log space is actually less dispersed than a density that ranges from -2 to -2.5. Additionally, some of the shape of the distributions is a result of using KDEs to plot the densities; we will be sure to include the sampled values along the x-axis, and make a clarifying note regarding the relative spread of the distributions in the camera ready.
>
> - For the setup of the CIFAR and UCI experiments we use one consistent architecture for all the UCI data and another for the CIFAR data. For the UCI data we use a small network with 3 hidden layers and 32 units in each layer, and for CIFAR we use an implementation of Deep-Conv from Towards Learning Convolutions from Scratch, Neyshabur, 2020. We will make this clarification explicit in the camera ready. It is also worth noting that we are not intending to draw comparisons between a single model’s performance on different data sources, but instead to highlight the consistency of RPP in cases where we either do or do not have reasons to favor the use of symmetries.
> -Although Hopper appears to have an exact front/back symmetry, the symmetry of the RL environment is actually broken by the fact that the ankle joint is not centered on the foot of the robot, and also that the reward of the RL environment points directionally to the forward direction. Since RPP is applied to the value function, it needs to model this directional reward and a model with exact symmetry would be unable to distinguish between moving towards and away from the goal.
> -For the model free RL experiments (figure 5), in early testing we found that using a raw EMLP for the value function tends to lead to bad outcomes with performance that is considerably worse than the baseline and sometimes the agent fails to learn at all. In cases like hopper where the reward directly violates the symmetry this is understandable.
> For the learning transition models for model based RL (table 3), we ran a new set of 3 trials with an raw EMLP transition model for each of the tasks and found the following results:
> | Rollout  |   Swimmer-v2 |   Hopper-v2 |     Ant-v2 |
> |:---------------------|-------------:|------------:|-----------:|
> | 10 Steps |    0.63±.08   |   1.35±.09   |  5.78±.08   |
> | 30 Steps |    2.0±.3   |   4.7±.4   | 15.7±.5    |
> |100 Steps |    5.0±.8   |   9.5±.8   | 22.9±1.6    |
>
>   The results are a little worse than the baseline MLP results from the paper, but not dramatically so. However, this performance is consistent with challenges of exploiting approximate symmetries with exactly equivariant models. We will add these results to the paper.
> Thank you for the note on the broader impacts statement. We will move this to the main text in the camera ready.
>
> In response to your more specific questions:
> - The prior is indeed on the magnitude, or more precisely the norm.
> - Good catch, A and B should be swapped in this sentence.
> - RPP-EMLP destabilizes SAC for Humanoid: due to using both value bootstrapping and off-policy data, SAC is notoriously sensitive to changes in the critic, which can often lead to runaway behavior from which the algorithm never recovers. EMLP (and therefore also RPP-EMLP) is less well conditioned than an ordinary MLP because of its bilinear layer. For instabilities in swimmer and walker we overcome this by lowering the critic moving average factor Tau (as detailed in table 4), but not for Humanoid. We don’t feel this is a fundamental limitation however, and it may be addressed with a more stable SAC algorithm, by applying RPP to a better conditioned equivariant network, or by using action, state, or reward normalization.

---

> > ### Comment · Reviewer_hyfb · 2021-08-30
> > **Thanks - maintaining my score**
> >
> >
> > Dear authors, thank you for your thorough response. It helps clear up some of the questions/concerns I had. I think addressing these in a revision will greatly improve the paper.
> >
> > I am inclined to maintain my (already very positive) score because the main weaknesses I see (limited methodological contribution and experimental evaluation) remain.

---

### Official Review · Reviewer_RJms · 2021-07-17

**Rating:** 7
**Confidence:** 4

**Summary:**

The idea here is to use a combination of fully connected layers with equivariant layers and use a prior to bias the feedforward layer towards simpler (equivariant) linear maps. Standard residual connections are given as an extreme example where the simpler linear map is the identity operator. L2 penalty on the weights is used to implement this prior. Experiments study this type of model in settings where 1) the exact symmetry holds, 2)  it is partially violated, and 3) the data has no such symmetry. Example application domains used for experiments include physical models and applications in RL, where the correct (approximate) symmetries are known.


**Limitations And Societal Impact:**

The appendix includes some discussion on the limitations.

**Main Review:**

The proposed idea is quite simple yet it seems quite powerful and flexible, in the sense that the model performs well even with misspecified or “approximate” symmetries. The experiments are chosen very well, and the results are supportive. The presentation is quite clear, the paper is an enjoyable read. I have a few (minor) questions:

The material in appendix D about the effect of prior variance seems quite central to your claim. I suggest expanding the experiments to see if similar patter (of robustness to the hyperparameters) appears across different problems and potentially add this to the main body.

Could you please elaborate on the claim of lines  165-168? Are these only based on empirical observations?

How do you explain the RL experiments in which RPP fails (Hopper-v2, Ant-v2)?

I wonder if presenting this idea as a generalization of residual connection is valid, given that a vanilla residual connection does not have an associated parameter.

Is it necessary to package this idea with the EMLP’s method for finding equivariant linear maps, given that one could use any other method for designing equivariant kernels or tying the parameters in the feedforward layer?


**Time Spent Reviewing:**

5

---

> ### Author Response · Authors · 2021-08-10
> **Author Response**
>
> Thank you for your thoughtful and supportive review! Addressing your question, the simplest choice in RPP of placing an independent Gaussian prior on an equivariant linear layer and the more general linear layer is equivalent to a prior with higher variance in the equivariant subspace than its orthogonal complement, regardless of scale of the two priors. This equivalence is true mathematically and not just empirically. Since equivariant solutions lie in a subspace of a larger vector space of more general linear layers, one can consider the more general layer as the sum of its projection to the equivariant subspace, and the orthogonal complement. Therefore the equivariant subspace gets the contribution from the prior variance of both layers, while the non equivariant orthogonal complement has the variance only from the general linear layer.
>
> We will expand on the material in Appendix D and shift more focus to this result in the camera ready. We performed a similar experiment to the one given in the paper on a further set of tasks including training on CIFAR-10 using weight decays on the equivariant and fully connected weights ranging from 1e-4 to 0.1. The result is similar to the one given in Figure 7, with the dominant trend being that lower precision on the convolutional weights leads to more accurate solutions. At the lowest precision level on the convolutional weights the performance of all models is similar, and at higher precision levels on the convolutional weights models with smaller precision on the fully connected weights perform better.
>
> While RPP applied to the value function performs well in the other 5 environments, it fails on the most challenging Humanoid task. In this nonlinear, off policy, with value bootstrapping setting of SAC the value function is fairly unstable and if it diverges the agent will never recover. The EMLP type networks tend to be slightly less well conditioned than their MLP counterparts on account of their bilinear layers and for Humanoid the value function never recovers. We think it may be possible to address this problem with better state, action, or reward normalization. However, in Figure 6 in appendix B we investigate applying RPP-EMLP to the policy function instead of the value function, and the training remains stable (SAC stability tends to be much more sensitive to the value function than the policy). For the model based experiment on Ant-v2, we are not sure why RPP performs worse.
>
> Can RPP be used with equivariant kernels and models other than EMLP? Absolutely! We choose to focus on the EMLP because of its generality, and ability to handle the diverse state and action representations (listed in appendix E) necessary for the RL experiments. However, the idea itself is broadly applicable to equivariant networks, and we thus expect it could have a significant impact in general. For other applications, other equivariant networks could be suitable than EMLP and we could use the RPP there in precisely the same way, as we do in Section 5.3 with RPPs and convolutional structure. We will emphasize this in the text.

---

### Official Review · Reviewer_wXcB · 2021-07-19

**Rating:** 5
**Confidence:** 3

**Summary:**

The authors propose to enforce soft equivariance constraints in ANN layers through a Gaussian prior on layer weights. Experimental results on a variety of problems support the benefit of the approach perhaps unsurprisingly. The overall contribution feels quite straightforward and incremental.

**Limitations And Societal Impact:**

I could not find a discussion on the limitations (e.g. computational complexity).

**Main Review:**

Overall the contribution of the paper is to use MAP optimization to propose and intermediate between stricly equivariant layers and non-equivariant ones (MLPs typically). This is enforced by using a Gaussian prior whose convariance is a sum of unstructured (proportional to identity) and structured (contrained to a basis of equivariant matrices) matrices.
Perhaps unsurprisingly, the resulting model achieves a trade off between unstructured approches and stricly equivariant approaches that is beneficial in several examples.

The contribution feels thus quite iterative, while there would be many ways to address the problem more in depth: e.g. can we use sparser priors in ordre to relax equivariance only where/when necessary? Can we implement this soft constraint such that computational benefits of structued models are preserved as much as possible?



**Time Spent Reviewing:**

3 hours

---

> ### Author Response · Authors · 2021-08-10
> **Author Response**
>
> Thank you for your comments and thoughtful review. We want to emphasize that the paper is making a significant contribution --- relaxing the fundamental tension between building expressive models, and incorporating useful inductive biases, coupled with strong results. Indeed, we show that for fully constrained or unconstrained settings, the RPP is competitive with perfectly specified models, and in cases where symmetries are not exact, the RPP is superior. As noted by other reviewers, we view the simplicity of the approach as a strength, providing an elegant mechanism for tackling a fundamental modelling challenge, with good results.
>
> In our understanding, your concern is that perhaps the approach is too iterative. In our opinion, although straightforward, RPP is not iterative. RPP addresses a significant limitation of equivariant networks (that they cannot fit data with approximate symmetries) with an approach that is broadly applicable to equivariant networks. While several previous approaches considered the need for symmetry breaking in networks as a burdensome detail with network specific remedies, we identify and address this problem directly. Your suggestion about exploiting sparse priors for computational benefit is interesting and may be quite beneficial in some settings. However, with the applications we explore in the paper the heterogeneous input and output representations would make it hard to find significant speedups over the dense implementation. With EMLP, there would be no computational gains for doing so since the equivariant layers are not any faster than generic fully connected layers, and this is often the case such as with GCNNs not being any more computationally efficient than standard CNNs.
> We would appreciate it if you would consider raising your score in light of our response.

---

> > ### Comment · Reviewer_wXcB · 2021-09-03
> > **Post-rebuttal and post-discussion feedback**
> >
> > Thank you for your reply. I will remain with my score (with intermediate confidence), because there is, in my view, not enough demonstration of the benefits of the approach for this type of empirical contribution.
> >
> > One reason why I think the current validation is insufficient is that it is quite expected that if you take two extreme sizes of model class (fully unconstrainted MLPs and very constrainted EMLP), a better solution lies in between. This could be expected from any other regularization, so one needs to go a little further to demonstrate the practical benefit of the idea (given that this is the only potential contribution of the paper that I see).
> >
> > Equivariance is only a means to an aim, and the aim is arguably incorporating assumed invariances of the dataset as an inductive bias (I believe you agree with this statement based on your introduction, although in my view you could frame the general goal of this research direction more explicitly/mathematically, notably to help the reader understand the broader context).
> > As I understand it, the equivariances included in the experiments of the present manuscript are based on invariance assumptions on the dataset, which thus opens the way to exploiting other baselines (such as augmentations).
> > For example this neurips paper: https://proceedings.neurips.cc/paper/2019/hash/1d01bd2e16f57892f0954902899f0692-Abstract.html does provide a quite general regularization framework that is compared to equivariant neural networks. Overall, justification for the benefits with respect to previous work (and notably restricting the comparison the EMLP and MLP) that you give in the introduction and the above rebuttal are, in my view, a relatively narrow set of the related literature that aims at enforcing inductive biases towards assumed symmetries of the datasets. Including a convincingly competitive baseline would certainly improve the paper, as well as framing the problem in a broader context.

---

> > > ### Author Response · Authors · 2021-09-03
> > > **Author Response**
> > >
> > > We appreciate the response, but we respectfully disagree with the assessment. Although you may not find the strong performance of RPP surprising, we don’t see this as a drawback. Existing equivariant architectures can’t accommodate approximate symmetries and RPP can, and it benefits from doing so. Additionally, developing equivariant methods for the continuous control RL tasks is itself a novel and non-trivial contribution because of the heterogeneous state and action spaces. We note that unlike architectural equivariance inductive biases, data based regularization approaches like data augmentations are less capable of preserving symmetries outside of the training examples as discussed below. Moreover, the correct way of applying these regularization methods to problems that require _equivariant_ outputs rather than invariant ones is open to interpretation and may require substantial investigation on its own.
> > >
> > > With regard to strong baselines, we would like to emphasize that in the reinforcement learning context, MLPs for the policy, value function, and transition models are the methods of choice when learning directly from the state variables and SAC represents a state of the art model free RL algorithm. These are the strongest baselines for the Mujoco locomotion tasks, and what we compare against. Unlike in other domains like images and text where models with well specified inductive biases like CNNs are widely used, the networks used in model free RL are unstructured, and this is precisely a result of the difficulty of incorporating equivariance for these heterogeneous objects and the fact the symmetries are not exactly satisfied. In fact, we tried using fully equivariant networks on these problems (see our response to reviewer 3), but these networks fail because of the symmetry breaking required to solve the environment.
> > >
> > > Other methods that enforce equivariance, such as data augmentation and consistency regularization referenced in the mentioned paper (https://proceedings.neurips.cc/paper/2019/hash/1d01bd2e16f57892f0954902899f0692-Abstract.html), suffer from many of the same problems. Data based regularization and augmentation do not constrain the model in regions of the input space that are sparsely covered in the training data, whereas architectural equivariance inductive biases do. In general, architectural inductive biases are better suited to capitalize on symmetries, and this is demonstrated specifically in the RL context in van der Pol et al, 2020. Additionally, for tackling the Mujoco problems we need equivariant outputs rather than invariant ones, and regularization methods in the mentioned paper have not been designed to accommodate this setting.
> > >
> > > Lastly, while our results are somewhat focused on RL problems, we do explore the use of RPPs in other problem settings. Table 1 shows that RPPs work as expected when combining MLPs and CNNs, showcasing that RPPs are not just an iteration on EMLP and have broad applicability in any setting where one aims to balance flexibility and strong inductive biases.
> > >
> > > Reference:
> > > Elise van der Pol, Daniel Worrall, Herke van Hoof, Frans Oliehoek, and Max Welling. Mdp
> > > homomorphic networks: Group symmetries in reinforcement learning. Advances in Neural
> > > Information Processing Systems, 33, 2020. https://arxiv.org/abs/2006.16908

---

### Author Response · Authors · 2021-08-10
**Overall Response**

Thank you for thoughtful and supportive reviews. We appreciate the reviewers for acknowledging our work as both well-motivated and well-executed. We believe residual pathway priors provide a clear solution to the timely problem of how to construct models with equivariant inductive biases but no hard constraints. In general, the question of how to effectively relax the tension between flexibility and inductive biases is pressing, and the RPP shows that this ambition can be achieved with a direct approach that has compelling empirical performance. In particular, we retain the strong inductive biases of architecture constraints when our symmetry assumptions are upheld, without being limited by them when they are violated, and can benefit from them when symmetries are approximate.

In addressing the reviews we have clarified and expanded a number of points in the paper, including: adding a comparison to exactly equivariant models on the RL tasks, expanding our investigation of the effect of prior variance to include RPP-conv models, and providing a better explanation of the symmetries present in the RL experiments and why RPPs are not equally performant across tasks. We have responded to each reviewer’s questions and comments in reply posts. We hope that our responses can be considered in the final evaluation for this paper.

---

### Public Comment · ~S_Chandra_Mouli1 · 2021-12-10
**A related work**

I enjoyed reading this paper, especially the use of equivariant bases and the orthogonal complement to have soft equivariances. In our work at ICLR 2021 (https://openreview.net/forum?id=7t1FcJUWhi3),  we learn which of the $m$ given groups a model should be invariant to (i.e., soft invariance). We use invariant bases for every group and their orthogonal complements to construct subspaces with varying degrees of invariance. Finally, we learn a model with maximum invariance that does not hinder training performance.

While we only consider invariance and finite groups, it is interesting to see in your paper that a similar approach can be used for equivariances and Lie groups as well. It would be nice if you could discuss these parallels in your paper so that the literature on learnable invariances/equivariances is not fragmented. Thanks!

---

### Decision · Program_Chairs · 2021-09-27

**Decision:**

Accept (Poster)

**Comment:**

This paper presents a simple solution to an important problem - deciding what kind of inductive biases to use when we don’t have full confidence about the type of symmetries that exist in the domain.  The solution is to combine a flexible model with one that has strong equivariance inductive biases.

All reviewers recognize the importance of the problem, clarity of the presentation and the simplicity of the approach.  The proposed method being simple does not diminish its value, and simple methods are typically more applicable to a wide range of problems and more easily adopted by many.

Reviewer wXcB who gave a 5 rating does have a point about the baselines though: this paper mostly contrasts the proposed approach with the two extremes - the MLP without any explicitly symmetry built in and models like convnets that have strong equivariance built in.  It would make the paper stronger to have a stronger baseline, that is for example trained with data-augmentation as a soft way to build some inductive bias into the model.  The advantage of the proposed approach, as the authors explained in the discussion, is that it will be much more data-efficient and generalize much better in data-sparse regions.

After discussion among the reviewers and calibrating across a range of borderline papers we decided to accept this paper based on its clarity and elegance, and believe it can be a useful contribution to the community.